# Germination and seedling establishment for hydroponics: The benefit of slant boards

**Noah James Langenfeld** (ID) *, **Bruce Bugbee**

Crop Physiology Laboratory, Department of Plants, Soils, and Climate, Utah State University, Logan, Utah, United States of America

* noah.langenfeld@usu.edu

## Abstract

Germination and seedling establishment for transplanting into hydroponics often uses porous substrates, but fine roots grow into these substrates, and they cannot be removed without damaging these roots. Seedlings transplanted without removal of substrates can cause interactions with solution chemistry or addition of particulates to the nutrient solution. Germination of seeds on slant boards is clean, uniform, and reduces the time to transplanting. Slant boards facilitate development of long roots, which maximize exposure of the primary root to the nutrient solution after transplanting. The "boards" are made from thin acrylic or polycarbonate sheets with germination paper on top. Seeds are held in place by covering with thin paper before vertical placement of the boards in the container. Four to twelve days later, the seedlings with long roots can be removed from the paper without damage and transplanted into the hydroponic system. Here we describe slant board construction and procedures for rapid germination and transplanting in hydroponics.

**Data Availability Statement:** All relevant data are within the paper and its Supporting Information files.

## Introduction

Hydroponics is often used as a research tool in plant nutrition studies because of the precise control of root-zone conditions and detailed monitoring of root health. Liquid culture hydroponics does not use a solid substrate and therefore has no adsorption and desorption of nutrients, which can complicate nutritional studies. These systems work well with the mass-balance approach as nutrients are either in the solution or in the plant [1].

Rapid, uniform germination is essential for research in hydroponic systems. Few seeds can germinate submerged in water [2] and thus cannot be planted directly into the nutrient solution in liquid hydroponics. Submersion in water inhibits oxygen diffusion across seed membranes and prevents germination [3].

For maximum uniformity, seeds should be germinated separately and transplanted into the main system when roots are long enough to be in contact with the nutrient solution immediately after transplanting (Fig 1). Reduced time for germination is beneficial so plants can be subjected to treatments in hydroponics systems for a larger percentage of their lifecycle.

**Funding:** This research was supported by the Utah Agricultural Experiment Station, Utah State University, and approved as journal paper number 9600; NASA, Center for the Utilization of Biological Engineering in Space (grant number NNX17AJ31G). The funders had and will not have a role in study design, data collection and analysis, decision to publish, or preparation of the manuscript.

**Competing interests:** The authors have declared that no competing interests exist.

Germination methods utilizing a solid substrate such as peat moss, coco coir, perlite, or vermiculite [4] are often used, but transplanting requires removing all the substrate from the roots to avoid its introduction into the hydroponic system. This process damages fine root hairs and stunts seedling growth following transplanting.

Seeds germinated in inert mineral wool or floral foam can be transplanted directly into hydroponic systems as plugs, but they can easily be flooded if the media contacts the solution after transplanting. Germinating on a rolled paper towel is common, but germination is not as uniform, and the density of the seeds is often excessive.

Transparent boxes with germination paper on the bottom are widely used for germination testing, but root and seedling growth are not straight, which makes transplanting difficult.

Germination in porous ceramic media (calcined clay) facilitates removal of roots from the substrate, but this media does not allow selection of plants with long roots, and it is time consuming to rinse the particles from the root surfaces without damaging the roots. This product was historically marketed under the trade name 'Arcillite®', but it is now sold under the trade name 'Profile® Greens Grade™' (https://www.profileevs.com/products/soil-amendments/profile-porous-ceramic-ppc). For many years it was used for plant growth studies on the International Space Station because the roots can be removed from the media for analysis [5].

Jones and Cobb [6] appear to be the first to use the slant board method to rapidly germinate seeds with long and straight roots. Their tests showed high germination rates for seeds on germination paper placed on an acrylic board at a 67° angle and covered with a single layer of cellulose tissue. We have adapted this method for the rapid germination and selection of the best seedlings for transplanting to hydroponic culture.

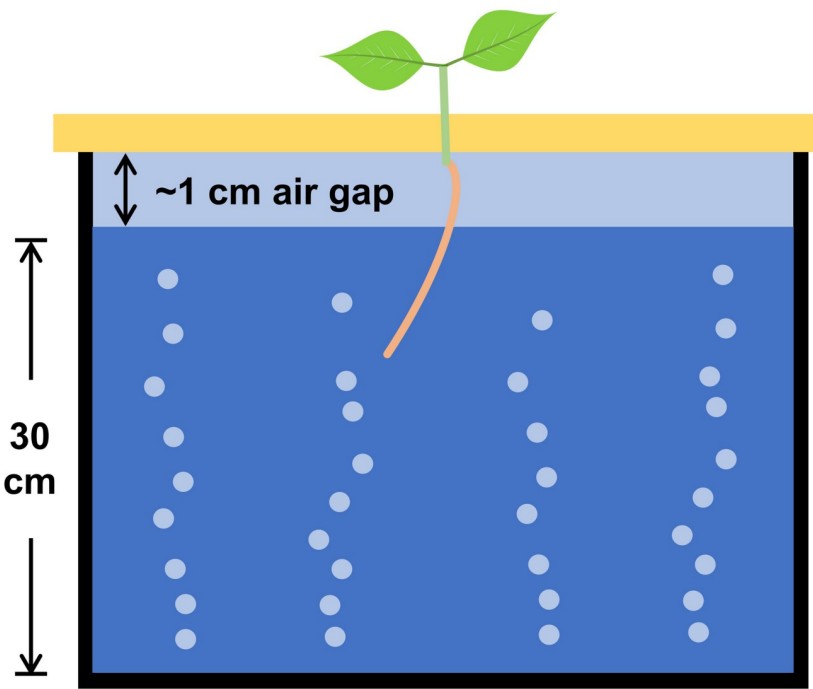

**Fig 1. Side-view diagram of seedling transplanted into a hydroponic tank.** The root must be long enough to span the 1 cm air gap and have ample access to the solution immediately after transplanting.

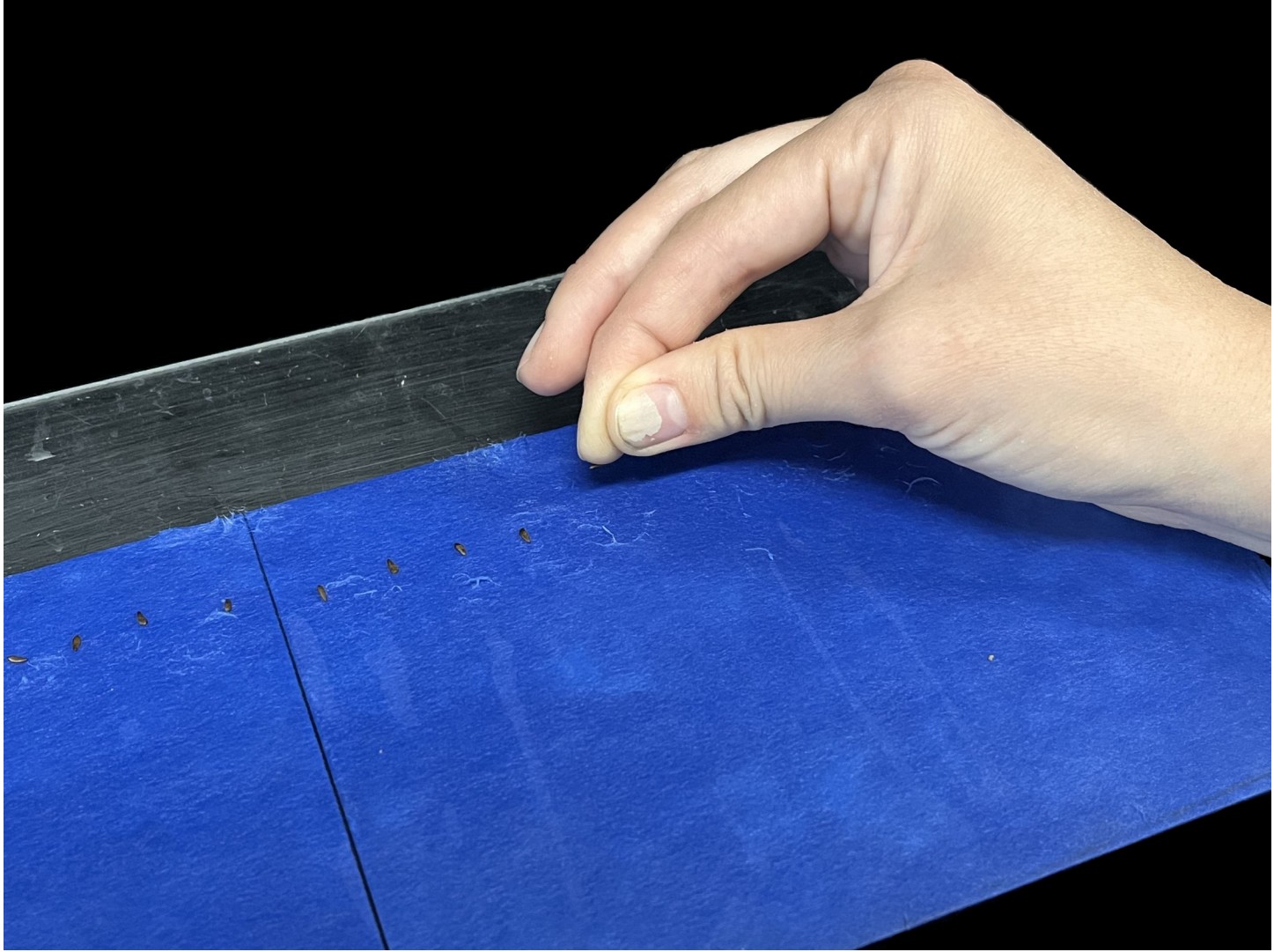

**Fig 2. Planting of lettuce (*Latuca sativa*) seeds on blue germination blotter paper.**

## Slant board system

A slant board system has five component parts: the board, germination paper, thin absorbent paper, an exterior container, and a positioning device.

The **slant board** is a thin (2 mm) piece of acrylic or polycarbonate cut to about 15 cm tall and 30 cm wide. Glass should be avoided due to the safety hazard from potential breakage and wood is not inert.

**Germination paper** is dense enough to prevent root penetration, has excellent hydraulic conductivity, is pH adjusted to a biological range (6–7), and is free from dyes that can influence root growth. It is sold by many suppliers (e.g., Seedburo, Anchor Paper, Nasco) and costs about 1 USD (2022) per 10 by 14-inch sheet. Paper color does not make a difference in germination quality, but fine root hairs are easier to monitor on blue colored germination paper [7] (Fig 2).

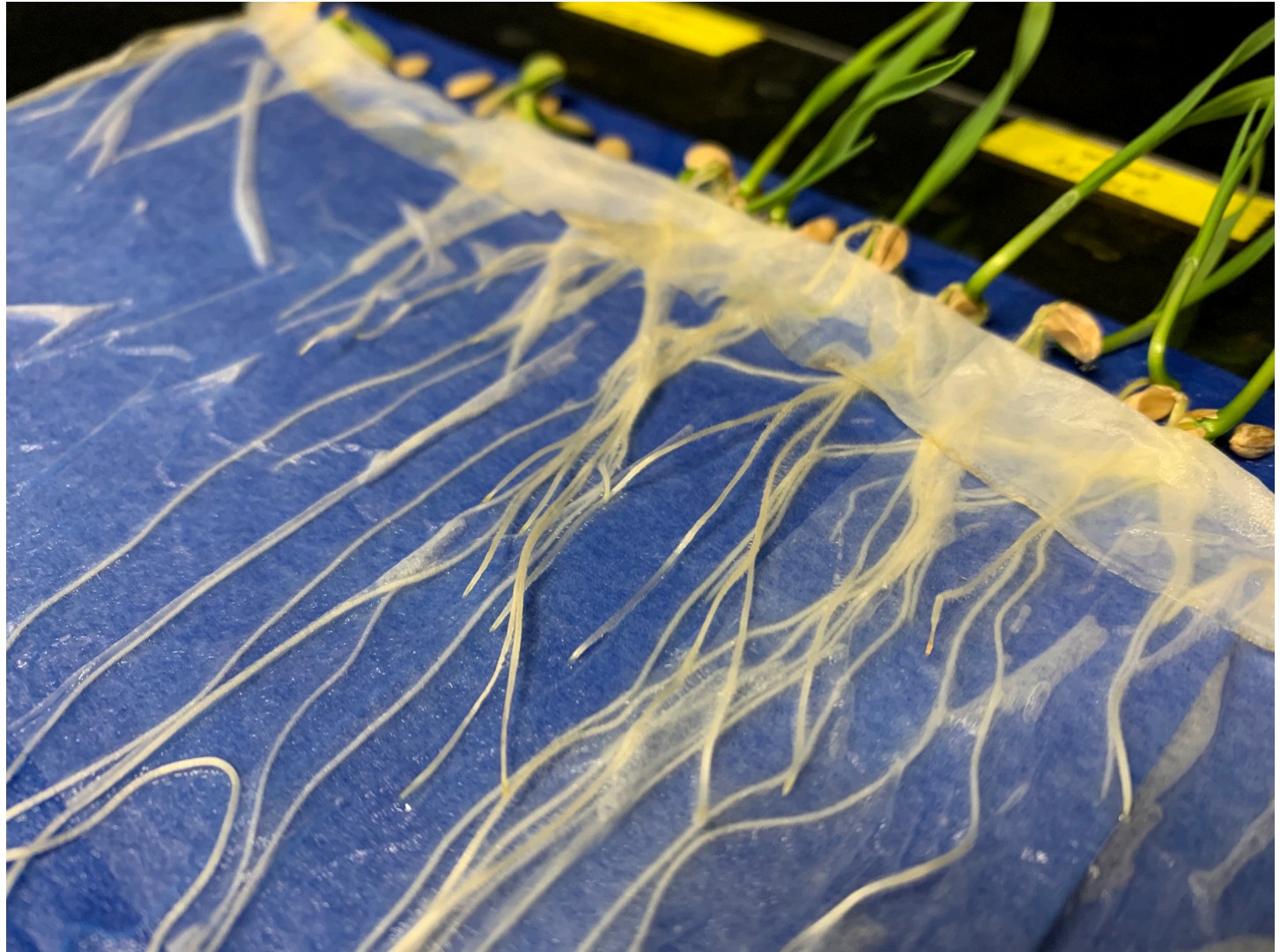

**Fig 3. Example of the growth of wheat roots through the thin absorbent paper.** It is difficult to separate the roots from the paper when this occurs. In this case, an additional piece of germination paper is recommended in place of the thin absorbent paper.

The **thin absorbent paper** is used to cover seeds. We have found single-ply 'Kimwipes' to work well because the roots are visible through the paper. Consumer-grade paper towels also work well and are widely available. Most roots do not penetrate the paper, but we have seen wheat roots grow through and become difficult to remove (Fig 3). A second piece of germination paper may be used in place of the thin paper in these situations. Tissue paper (e.g. Kleenex®) should not be used because it breaks when water is added to the system.

The **exterior container** can be any type of inert box that holds 2 to 3 cm of water and is large enough to hold all slant boards. Plastic boxes are lightweight, inert, and available in a variety of shapes and sizes. The container may be clear or opaque, but opaque containers may help limit direct light exposure to fragile roots.

The **positioning device** is used to hold the slant boards at angles of 70˚ – 80˚ in the exterior container and should be plastic to avoid corrosion. This can be milled to size with a table saw

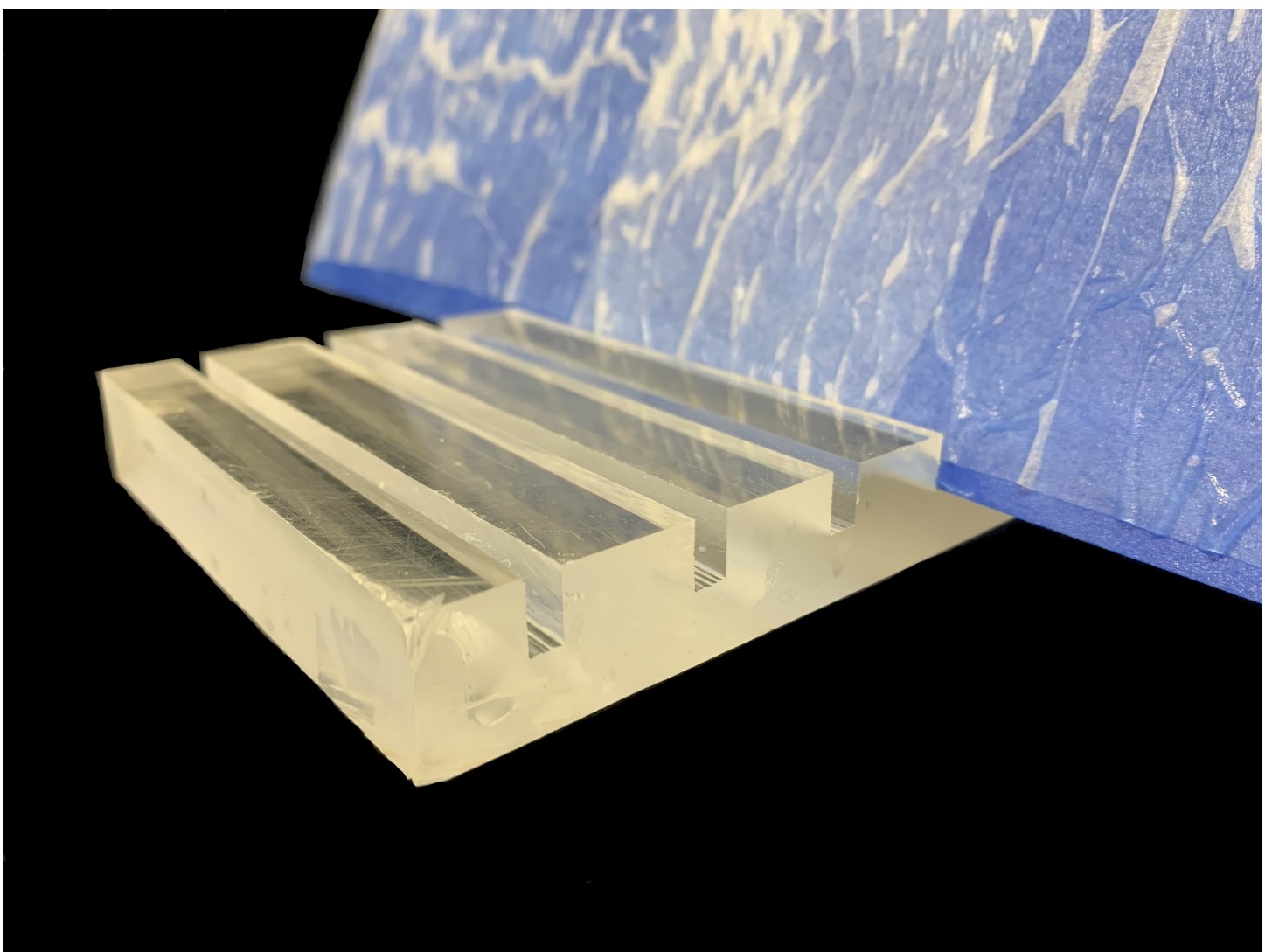

**Fig 4. Acrylic holder for slant boards.**

from a thick piece of acrylic (Fig 4) or polycarbonate. Aluminum (Al) should be avoided as it can release Al into the solution.

## Seedling variability and environmental conditions

More seeds should be planted than needed to ensure selection for uniform roots. We recommend overseeding by 100%. Root entanglement was not a problem at the high seeding density because the germinated seeds are quickly removed for transplanting. The cost of overseeding is outweighed by the ability to select for uniformity.

A complete nutrient solution is not necessary as seeds have enough nutrients for the first few days of growth and it quickly leads to undesirable algal growth on the surfaces exposed to light. However, water with about 40 ppm (1 mM) calcium (Ca) should be used for germination because Ca promotes healthy growth of seedling roots. Calcium nitrate can be added to purified water to achieve this Ca concentration. The water level in the plastic container should be maintained so that the level is about 2 cm above the bottom of the

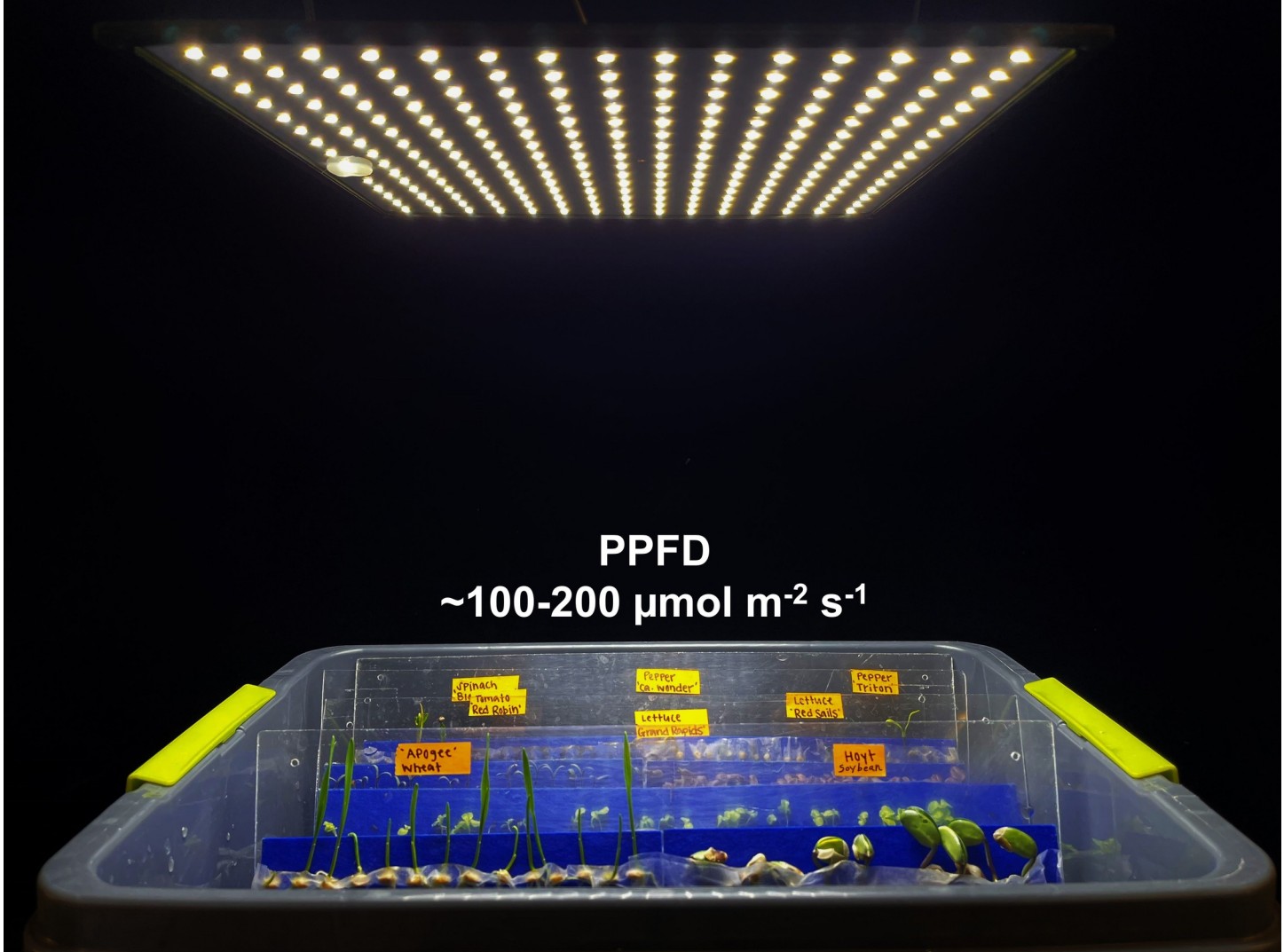

**Fig 5. Germination slant boards under LED lights.** The photosynthetic photon flux density should be about 100 to 200 $\mu mol\ m^{-2}\ s^{-1}$ to provide adequate light to reduce excessive hypocotyl extension.

germination paper. The need for water to maintain an optimal depth is determined by environmental conditions. Evaporation is less than 3 mm per day under electric lights and water can be added every few days. Levels lower than 2 cm may lead to desiccation and levels higher than 2 cm result in excessive water at the seed level and reduced oxygen transfer to the seeds.

Light is required for germination of some species, and it keeps hypocotyls short after the cotyledons emerge. Exposure to direct sunlight should be avoided, but indirect sunlight and supplemental lighting help with early photosynthesis of young seedlings. Light levels in laboratories are usually less than 20 $\mu mol\ m^{-2}\ s^{-1}$, which is inadequate to keep the seedling short and can lead to hypocotyl elongation [8]. We have used small, low-cost LED fixtures to provide 100 to 200 $\mu mol\ m^{-2}\ s^{-1}$ over slant boards on a laboratory bench (Fig 5). These are available from multiple sources (See examples in this home horticulture video: https://www.youtube.com/watch?v=m3rBpc7ue74).

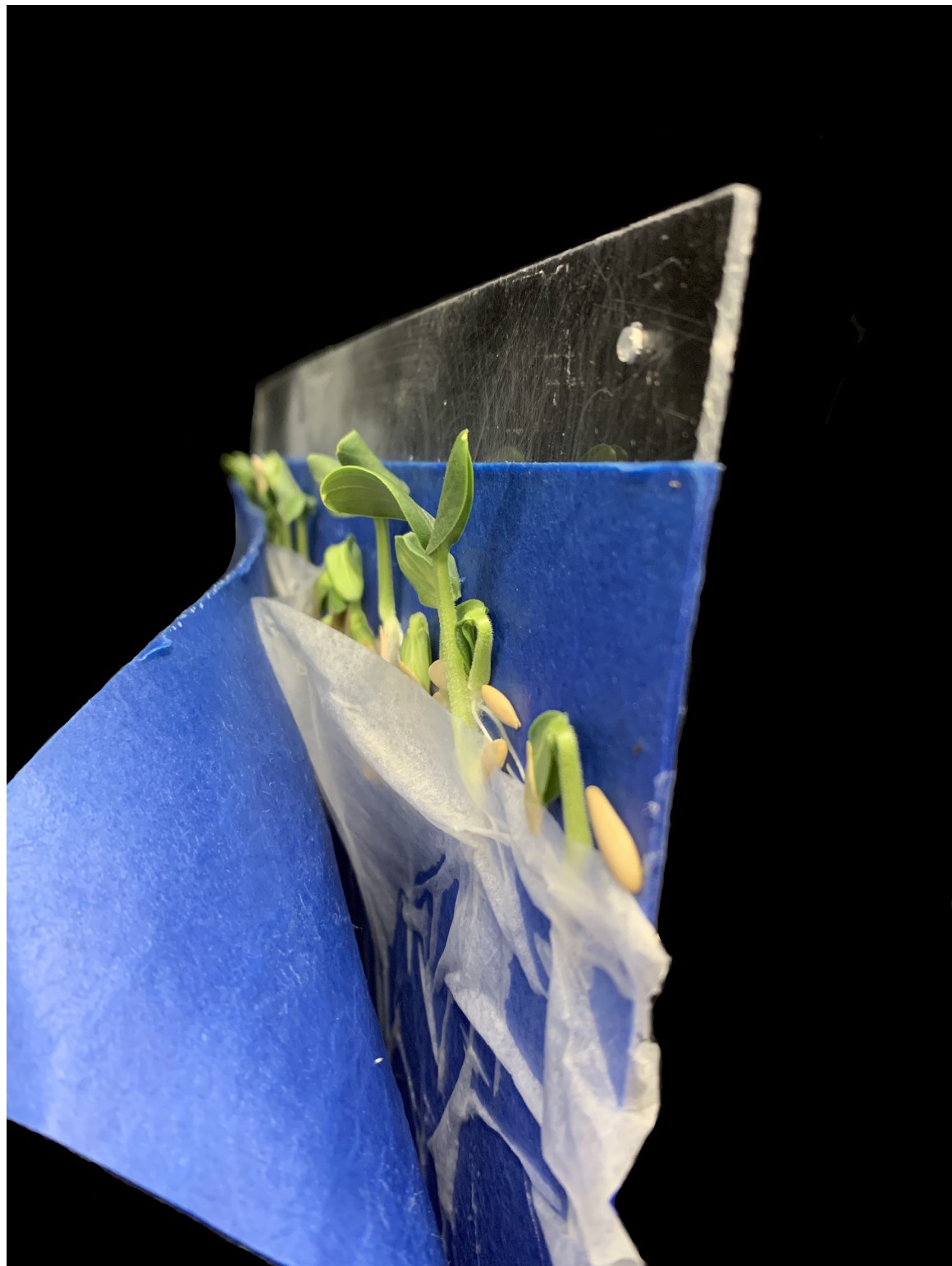

**Fig 6. Side view of cucumbers on slant board with an additional piece of germination paper on top of the thin absorbent paper.**

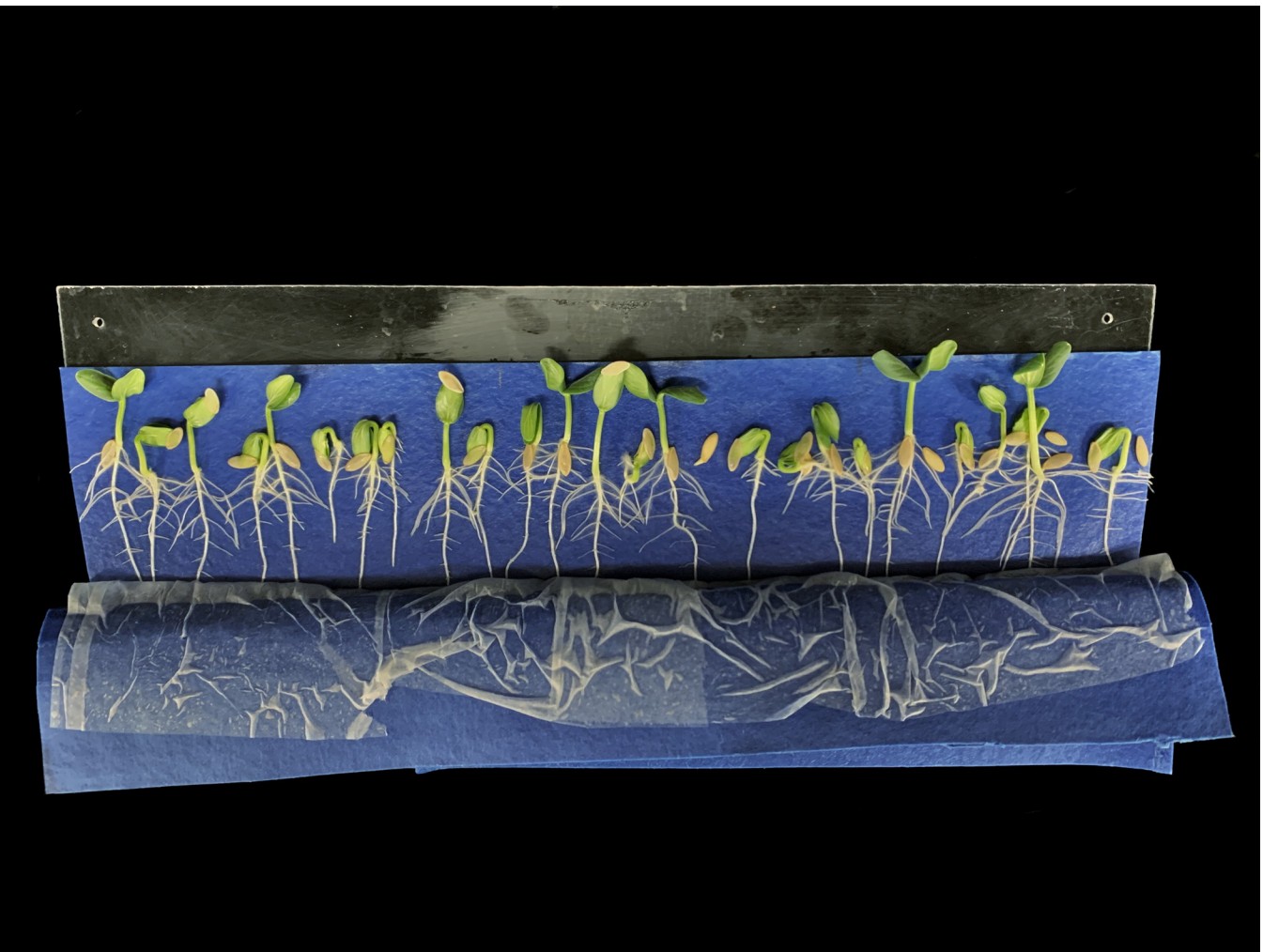

**Fig 7. Cucumber seedlings on a slant board with two layers of germination paper.** The roots of cucumber seedlings branch more than other species such as lettuce and tomato.

The temperature should be 20 to 25˚C. Higher temperatures can increase germination rates but decrease the solubility of oxygen in water and can decrease root elongation uniformity; lower temperatures often increase time to germination [9, 10].

Not all seeds germinate and grow at the same rate. We have found that lettuce (*Latuca sativa*), wheat (*Triticum aestivum*), and soybean (*Glycine max*) germinate quickly and are ready for transplanting in less than seven days. Tomato (*Solanum lycopersicum*) and pepper (*Capsicum annuum*) require more than seven days. Some species with large seeds, such as corn (*Zea mays*) and cucumber (*Cucumis sativus*), require additional moisture for germination. Placing a second piece of germination paper on top of the seeds can help to maintain moisture for these species (Figs 6 and 7).

Spinach (*Spinacia oleracea*) is difficult to germinate because of a hard seed coat and germination inhibitors [11]. Following the method of Katzman et al. [12], we significantly improved germination with a 4-hr pretreatment in a continuously-stirred solution of 0.5% sodium hypochlorite (a 1:10 dilution of standard commercial bleach), followed by a 15-hr rinse in continuously aerated distilled water to remove bleach residue. Seeds were then placed in germination

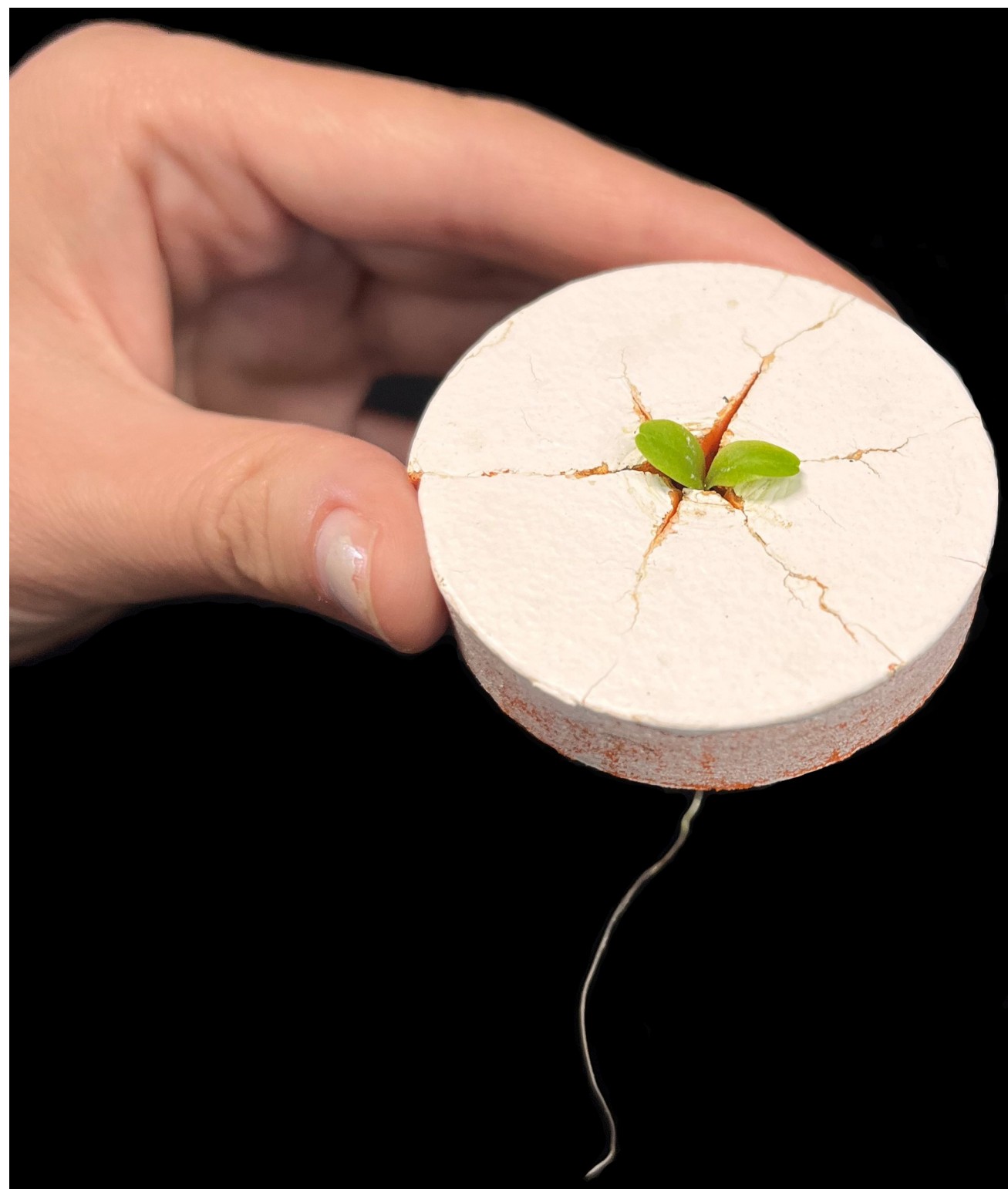

**Fig 8. Neoprene cloning collar with lettuce seedling.** The top of the collar has been painted white to reduce light penetration into the nutrient solution, reduce temperature, and to reflect light back to leaves for maximum growth. A radial slit in the neoprene collar runs between the thumb and the plant.

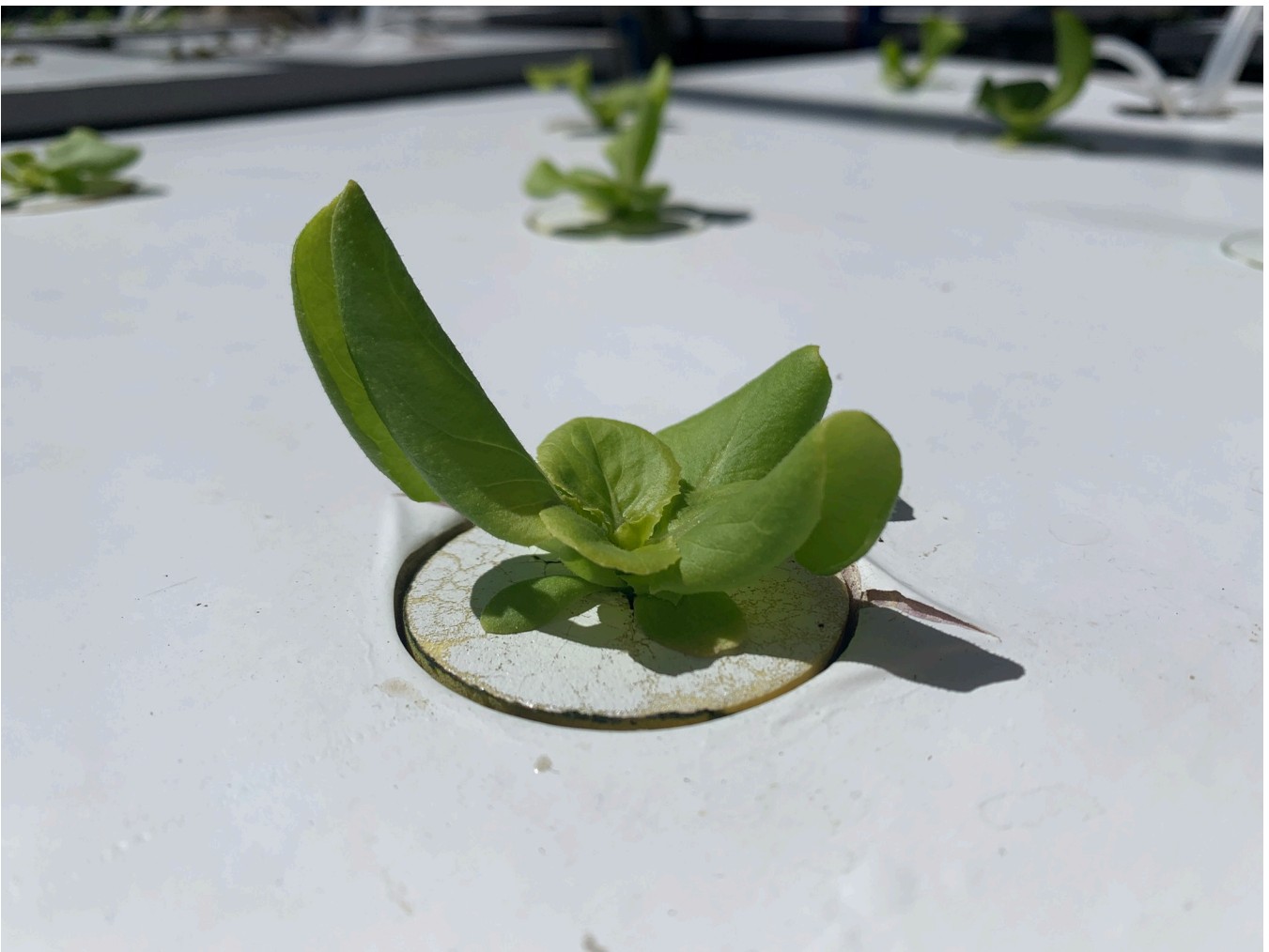

**Fig 9. Lettuce seedling in neoprene collar placed in rigid cover above nutrient solution surface about 7 days after transplanting.**

boxes with germination paper soaked with 0.3% hydrogen peroxide (a 1:10 dilution of standard commercial hydrogen peroxide) to signal germination. Both bleach and hydrogen peroxide are beneficial for maximum germination. It was also reported that lowering temperature from 30 to 18°C increased germination. We have used this method to increase germination of 'Bloomsdale' and 'Melody' spinach cultivars from 20 to over 90% compared to non-treated controls. Wojtyla et al. [13] reviewed the value of hydrogen peroxide as a signaling molecule in promoting germination. As soon as the radicle emerges, the seeds should be transferred to slant boards for root elongation.

## Transplanting into hydroponic systems

Seeds should be transplanted as soon as the roots are adequately long. Growth rate is diminished if seedlings are left on the slant board because they do not have access to nutrients. Root elongation starts to decrease after about seven days for lettuce and 12 days for tomato.

Neoprene cloning collars are used to hold the plants above the solution. These are widely sold online (Amazon.com, Walmart.com). These collars come with a radial slit in one side, gently hold young seedlings above the nutrient solution, and minimize root growth into the

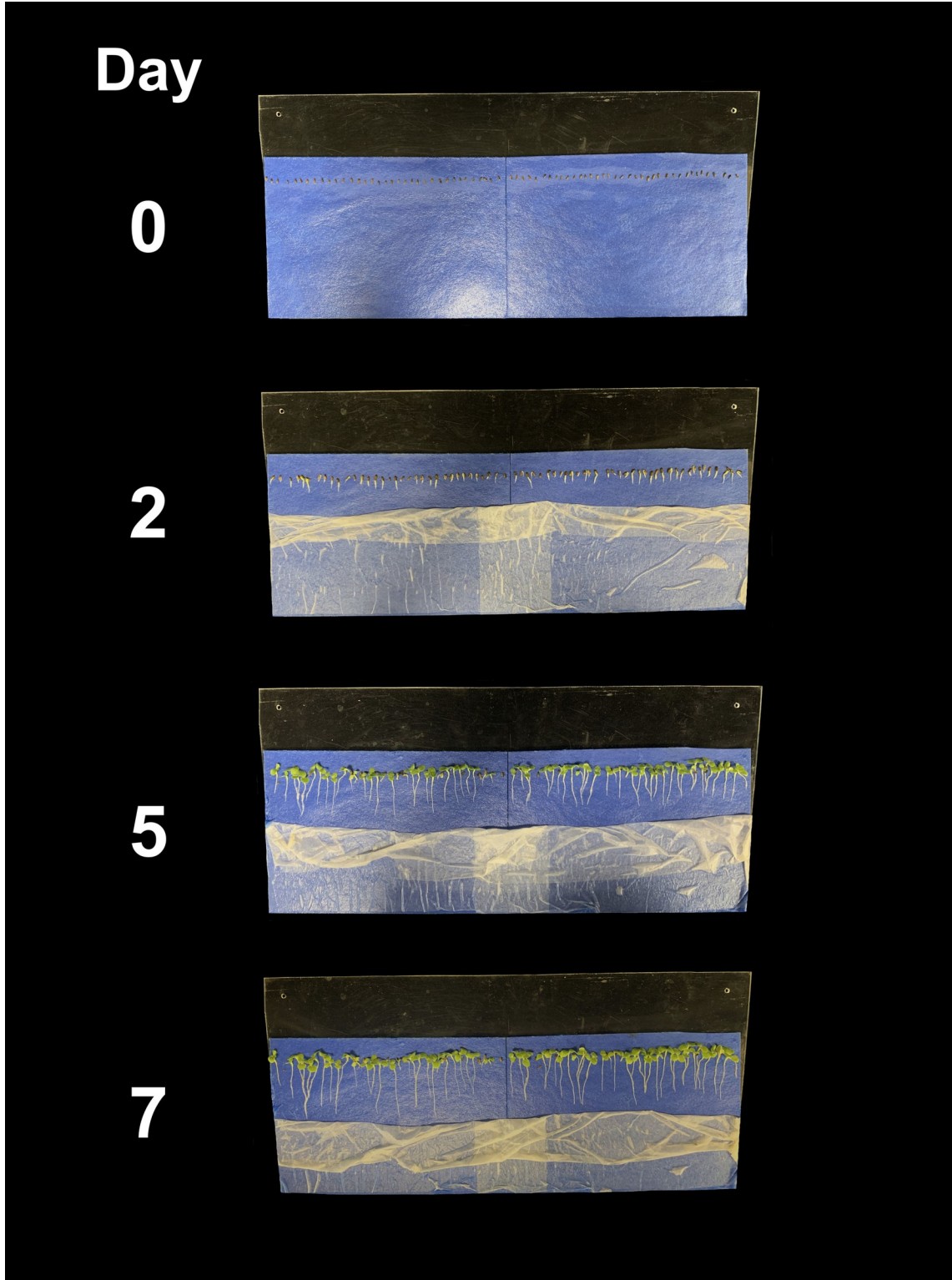

**Fig 10. Slant boards with germination blotter paper, lettuce seedlings, and thin paper over the seeds from day zero to day seven post seeding.** The thin paper can be removed to expose roots for inspection. Over-seeding helps for selection of uniform roots.

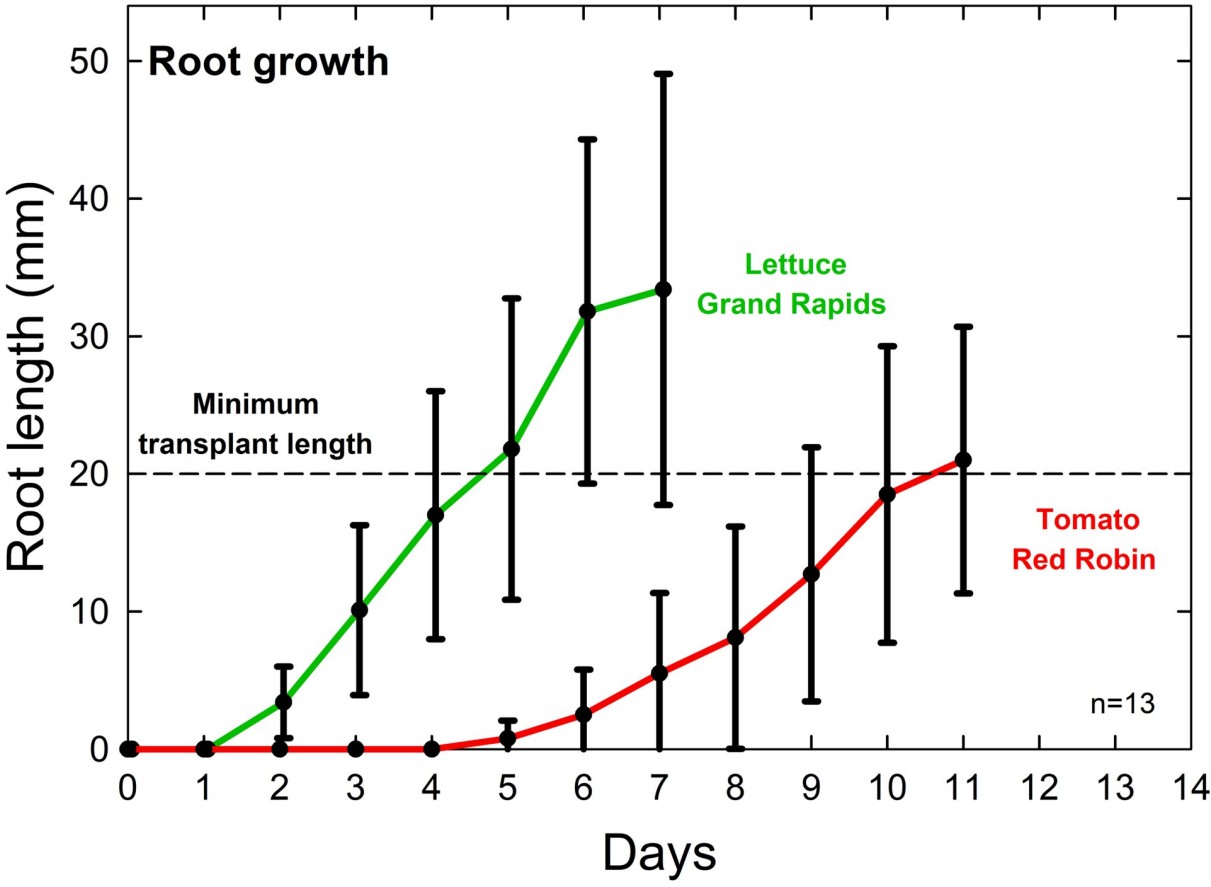

**Fig 11. Example of rapid and excessive growth of lettuce (cv. Grand Rapids) and tomato (cv. Red Robin) roots over time.** Root elongation rate starts to decrease after roots are 20 to 30 mm long. Error bars represent standard deviation, n = 13. There is a large variability among seedlings. Overplanting by a factor of at least two allows for selection of the most vigorous seedlings.

material (Fig 8). These collars can be directly placed in a rigid cover (Fig 9) or can be floated on the nutrient solution surface.

Germinating seeds on a slant board facilitates rapid growth of long, straight roots. Uniform seedlings can be selected and easily removed for transplanting without damaging root hairs. All components except the papers are reusable and the system is easily scalable for multiple seed sizes and species.

## Materials and methods

The protocol described in this peer-reviewed article is published on protocols.io, https://doi.org/10.17504/protocols.io.5jyl89b86v2w/v2 and is included for printing as S1 File with this article.

## Expected results

Use of a slant board results in rapid root elongation and easy transplanting four to seven days following germination for lettuce (Fig 10) and nine to 11 days after germination for tomato. Seedling vigor is reduced after this time as nutrients stored in plant seeds get depleted. Transplanting into nutrient solution should occur around day five for lettuce and day 10 for tomato to maintain exponential crop growth (Fig 11). Lettuce that is transplanted into treatments

seven days after seeding and is harvested 35 days after seeding means 80% of the lifecycle occurs when exposed to treatment conditions.

## Supporting information

**S1 File. Protocol for slant board method from protocols.io.**
(PDF)

**S2 File. Root length measurements from Fig 11 in main text.**
(DOCX)

## Author Contributions

**Conceptualization:** Noah James Langenfeld.

**Investigation:** Noah James Langenfeld.

**Writing – original draft:** Noah James Langenfeld.

**Writing – review & editing:** Noah James Langenfeld, Bruce Bugbee.

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
