## [Decision Letter · Decision Letter 0]

12 Sep 2022

PONE-D-22-22878Germination and seedling establishment for deep-flow hydroponics: The benefit of slant boardsPLOS ONE

Dear Dr. Langenfeld,

Thank you for submitting your manuscript to PLOS ONE. After careful consideration, we feel that it has merit but does not fully meet PLOS ONE’s publication criteria as it currently stands. Therefore, we invite you to submit a revised version of the manuscript that addresses the points raised during the review process.

We look forward to receiving your revised manuscript.

Kind regards,

Meenakshi Thakur, Ph.D.

Academic Editor

PLOS ONE

Journal Requirements:

Additional Editor Comments:

Dear Dr. Noah James Langenfeld,

I can now inform you that the Academic Editor has evaluated the manuscript PONE-D-22-22878: Germination and seedling establishment for deep-flow hydroponics: The benefit of slant boards. I am pleased to inform you that it has been favourably received. You are advised that the manuscript will be acceptable subject to satisfactory minor revision. The comments below should be taken into account when revising the manuscript. Along with your revised manuscript, you will need to supply a covering letter in which you list all the changes you have made to the manuscript, and in which you detail your responses to all the comments passed by the reviewer(s) and me. Should you disagree with any comment(s), please explain why.

Academic Editor’s comments: Add additional information about the following points:

1. You recommend overseeding by 100% to ensure selection for uniform roots. Give the data about germination percentage. Did any entanglement observed between the roots. If yes, how did you separate entangled roots?

2. How frequently you added the water in the exterior container so as to maintain the level about 2 cm above the bottom of the germination paper?

3. Check the format of writing the references clearly.

Reviewer 1: Authors improve a method of old publication and applied to the contemporary hydroponics. The presented ideas are simple and reasonable, and this method is useful for various researchers. I think it is acceptable for publication in PLOS ONE.

I request to add information about experimental condition as described below. I think it will be helpful for users.

1. Compared to the size of slant board, blue-colored paper seemed to be smaller, and the position of seeds is lower than the edge of paper. How long?

2. The density of seeds is important to prevent the root from entanglement with neighbor plants. Do you have data or estimation about that?

3. Describe the protocol in sufficient detail.

Reviewer Recommendation Term: Minor Revision

Reviewer 2: There are few queries and suggestions listed in the attached edited reviewer's copy.

Reviewer Recommendation Term: Minor Revision

Reviewers' comments:

Reviewer's Responses to Questions

**Comments to the Author**

1. Does the manuscript report a protocol which is of utility to the research community and adds value to the published literature?

Reviewer #1: Yes

Reviewer #2: Yes

2. Has the protocol been described in sufficient detail?

Descriptions of methods and reagents contained in the step-by-step protocol should be reported in sufficient detail for another researcher to reproduce all experiments and analyses. The protocol should describe the appropriate controls, sample sizes and replication needed to ensure that the data are robust and reproducible.

Reviewer #1: Partly

Reviewer #2: Yes

3. Does the protocol describe a validated method?

Reviewer #1: Yes

Reviewer #2: Yes

4. If the manuscript contains new data, have the authors made this data fully available?

Reviewer #1: Yes

Reviewer #2: Yes

**5. Is the article presented in an intelligible fashion and written in standard English?**

Reviewer #1: Yes

Reviewer #2: Yes

6. Review Comments to the Author

Reviewer #1: Authors improve a method of old publication and applied to the contemporary hydroponics. The presented ideas are simple and reasonable, and this method is useful for various researchers. I think it is acceptable for publication in PLOS ONE.

I request to add information about experimental condition as described below. I think it will be helpful for users.

(1)Compared to the size of slant board, blue-colored paper seemed to be smaller, and the position of seeds is lower than the edge of paper. How long ?

(2)The density of seeds is important to prevent the root from entanglement with neighbor plants.

Do you have data or estimation about that ?

Reviewer #2: Dear Authors,

Greetings

There are few queries and suggestions listed in the attached edited reviewer's copy.

best wishes.

7. PLOS authors have the option to publish the peer review history of their article (what does this mean?). If published, this will include your full peer review and any attached files.

Reviewer #1: No

Reviewer #2: **Yes: **Dr Tanuja Rana

Assistant Professor

Deptt. Agricultural Biotechnology

College of Agriculture

CSK HPKV, Palampur 9HP)-176 061 India

---

## [Author Response · Author response to Decision Letter 0]

16 Sep 2022

Academic editor:

We added two sentences (line 103) describing how root entanglement was not observed regardless of the high planting density. This is an important consideration; thank you for your comment.

We added two sentences (line 111) discussing how frequently water must be added to maintain the 2 cm depth. Thank you for this comment.

References have been checked to ensure uniformity.

Reviewer 1:

The slant board is simply larger than the blue colored paper to leave room for labeling. The positioning of the seeds is described in step 4 of the protocol on protocols.io.

Two sentences have been added discussing seeding density and root entanglement. See line 111.

We believe the protocol is described in sufficient detail on protocols.io. Please see this for the detailed protocol.

Reviewer 2:

Thank you for your edits in the abstract. We have included most of these to make the abstract stronger. We also broaden the title and last sentence of the abstract to focus on hydroponics in general as opposed to specifically deep-flow.

Line 69 changed from plural to singular method.

We added two fundamental references describing the effects of temperature on seed germination and dissolved oxygen. Thank you for this comment.

We simplified the sentence about hydrogen peroxide alone being better than bleach alone.

We have fixed the missing Wojtyla reference.

We have changed our “We have…” statements to be more objective per your recommendation.

One line 183, numbers above 10 should be numbers and numbers below 10 should be spelled out. We edited the following sentence per your recommendation.

---

## [Editor Report · Decision Letter 1]

21 Sep 2022

Germination and seedling establishment for hydroponics: The benefit of slant boards

PONE-D-22-22878R1

Dear Dr. Noah James Langenfeld,

We’re pleased to inform you that your manuscript has been judged scientifically suitable for publication and will be formally accepted for publication once it meets all outstanding technical requirements.

Kind regards,

Meenakshi Thakur, Ph.D.

Academic Editor

PLOS ONE

Additional Editor Comments (optional):

Dear Dr. Noah James Langenfeld,

I have checked your revised manuscript carefully for the incorporation of changes suggested by me and reviewers. You have made most of the changes as suggested, I am pleased to tell you that your work has now been accepted for publication in PLOS ONE.
---

## [Editor Report · Acceptance letter]

26 Sep 2022

PONE-D-22-22878R1 

Germination and seedling establishment for hydroponics: The benefit of slant boards 

Dear Dr. Langenfeld:

I'm pleased to inform you that your manuscript has been deemed suitable for publication in PLOS ONE. Congratulations! Your manuscript is now with our production department. 

Kind regards, 

on behalf of

Dr. Meenakshi Thakur 

Academic Editor

PLOS ONE